# The Emerging Role of the Cancerous Inhibitor of Protein Phosphatase 2A in Pulmonary Diseases

**DOI:** 10.3390/medicina61101740

**Published:** 2025-09-25

**Authors:** Hamza Hamza, Dinesh Nirmal, Stephanie Pappas, Ugochukwu Ebubechukwu, Sunydip Gill, Adam Al-Ajam, Michael Ohlmeyer, Patrick Geraghty

**Affiliations:** 1Department of Medicine, Downstate Health Sciences University, State University of New York, 450 Clarkson Avenue, Brooklyn, NY 11203, USA; hamza.hamza@downstate.edu (H.H.); dinesh.nirmal@downstate.edu (D.N.); stephanie.pappas@downstate.edu (S.P.); ugochukwu.ebubechukwu@downstate.edu (U.E.); sunydip.gill@downstate.edu (S.G.); adjam800@gmail.com (A.A.-A.); 2Atux Iskay LLC, Plainsboro, NJ 08536, USA; michael.ohlmeyer@gmail.com

**Keywords:** cancerous inhibitor of PP2A, protein phosphatase 2A, pulmonary diseases, immune responses, cell cycle, signaling transduction

## Abstract

Promising protein targets are observed to play a role in multiple pathways across a variety of diseases, such as the regulation of immune responses, cell cycle, senescence, and DNA repair. The oncoprotein cancerous inhibitor of protein phosphatase 2A (CIP2A) can coordinate all these cell characteristics predominately by inhibiting the activity of the serine threonine protein phosphatase 2A (PP2A). CIP2A directly interacts with PP2A and other proteins, such as the DNA damage protein topoisomerase II-binding protein 1, to regulate signal transduction. CIP2A is overexpressed in many human cancers, including small and non-small cell lung cancers. High CIP2A expression in lung cancer correlates with poor prognosis, increased tumor proliferation, and resistance to targeted therapies or chemotherapy. Interestingly, CIP2A expression or signaling is also observed in several non-cancerous pulmonary diseases, such as chronic obstructive pulmonary disease. CIP2A can determine whether DNA-damaged cells enter mitosis and can mediate whether DNA repair occurs. CIP2A is also a regulator of inflammation and possibly fibrotic responses. Its functions are linked to altered NFκB activation and TNFα, IL-1β, IL-4, IL-6, IL-10, IL-13, and TGFβ signaling. This review outlines the possible impact of CIP2A-mediated signaling in pulmonary diseases, the processes that regulate CIP2A responses, CIP2A-dependent pathways, and potential therapeutic strategies targeting CIP2A. Substantial medicinal chemistry efforts are underway to develop therapeutics aimed at modulating CIP2A activity. The development of specific inhibitors of CIP2A that selectively target its expression or protein stability could improve our understanding of CIP2A’s function in pulmonary diseases.

## 1. Introduction

Cancerous inhibitor of PP2A (CIP2A) is an oncoprotein that is well documented as being a major player in cancer, primarily by inhibiting the activity of the serine/threonine phosphatase protein phosphatase 2A (PP2A) [1]. However, in recent years several research studies suggest that CIP2A may also play a role in pulmonary diseases, particularly in conditions driven by inflammation, fibrosis, and tumorigenesis [2,3,4,5]. Enhanced CIP2A expression is frequently overexpressed in non-small cell lung cancer (NSCLC) [6] and small cell lung cancer (SCLC) [7], where it promotes tumor progression. Elevated CIP2A expression is associated with poor prognosis, chemoresistance, and increased metastatic potential in lung cancer patients [6]. Altered CIP2A signaling is also reported in chronic obstructive pulmonary disease (COPD) [2,3] and recently in a model of bronchiolitis obliterans [4]. Several studies also suggest that CIP2A could play a role in fibrotic pathways and epithelial-to-mesenchymal transition (EMT) [4], which occur in idiopathic pulmonary fibrosis (IPF) or fibrosis secondary to chronic lung injuries. Equally, transforming growth factor beta (TGF-β) signaling, a key driver of pulmonary fibrosis, is linked to PP2A and CIP2A signaling [8,9]. Additionally, dysregulation of PP2A activity, possibly via CIP2A signaling, is linked to vascular remodeling, vascular smooth muscle proliferation, and endothelial dysfunction [10,11], which are central to the development of pulmonary hypertension. CIP2A is also a major regulator of immune responses, especially nuclear factor kappa B (NFκB) and cytokine pathways [4,12], which could make it relevant in diseases such as pneumonia, acute lung injury (ALI), and acute respiratory distress syndrome (ARDS).

### Global Impact of Respiratory Diseases

CIP2A overexpression is ubiquitous in different lines of cancer cells, with overexpression being present in 39–90% of tissue samples in various gastric, bladder, ovarian, tongue, hepatocellular, colon, NSCLC, and chronic myelogenous leukemia cells [13]. There is a positive correlation of CIP2A expression with tumor grade in many cancers [6,14]. However, altered CIP2A expressions are also observed in non-cancerous diseases. Before delving into the function and signaling responses of CIP2A, it is important to outline the relevance of the pulmonary diseases, in which CIP2A could contribute to initiation or progression.

Chronic respiratory diseases (CRD) constitute a significant global health challenge affecting almost 545 million individuals, who represent 7.4% of the world’s population [15,16]. CRDs like respiratory tract infections, asthma, COPD, pulmonary fibrosis, and lung cancers greatly contribute to the increasing costs of healthcare, accounting for about EUR 380 billion among 28 European nations in 2019 [15,17]. Lung cancer remains one of the most prevalent cancers and is the leading cause of cancer-related deaths worldwide, with NSCLC accounting for about 87% of all lung cancer cases and a 5-year survival rate of around 26%. SCLC makes up approximately 13% of cases, with a 5-year survival rate of less than 7% [18].

COPD is currently the fourth leading cause of death globally with approximately 3–4 million deaths per year, after the emergence of coronavirus disease 2019 (COVID-19) [19]. By 2020, COPD alone accounted for nearly 6% of the total direct annual healthcare expenditure in Europe, and almost USD 800 billion is projected in the next 20 years in the United States [15,20,21]. Smoking habits are implicated in more than 70% of COPD cases in high-income countries and are responsible for 8 out of 10 COPD-related deaths [19,22]. Other reported risk factors include household and outdoor air pollutants, genetics, occupational exposures, and respiratory infections [21]. The estimated incidence rates of IPF ranged from 3 to 9 per 100,000 population per year between 1998 and 2012 in Europe and North America [23]. However, once diagnosed, IPF carries mortality burdens like several cancers and yields a poor 5-year survival outcome [24].

Although pulmonary hypertension affects roughly 1% of the global population, its prevalence rises to 10% among individuals over 65 years old. The median survival for untreated patients is 2.8 years, although recent advancements in treatment have significantly improved the survival outcomes [25,26]. ALI and ARDS are acute respiratory conditions that can significantly impact morbidity and mortality, with ARDS affecting approximately 3 million people worldwide, accounting for approximately 10% of all intensive care unit admissions [27]. The mortality rates for ALI and ARDS can be as high as 30–50% in the United States [28,29]. Therefore, pulmonary diseases are a major obstacle, and determining new targets, such as CIP2A, is critical in identifying future therapies.

## 2. The Function of CIP2A

CIP2A, also known as p90, is a regulatory protein that is primarily known for its inhibition of the activity of PP2A, a phosphatase that is involved in many cellular functions, including cell cycle regulation, transductive signaling, metabolism, and responses to external stresses [30,31]. CIP2A also interacts with the DNA repair scaffold protein topoisomerase II binding protein 1 (TopBP1) to play major roles in the DNA damage response, the G2/M checkpoint, protection of chromosomal integrity during mitosis, and chromothripsis (complex patterns of chromosome rearrangements) [32,33,34]. CIP2A also plays a major role in the spermatogonial progenitor cell turnover and sperm counts by regulating expression of promyelocytic leukemia zinc finger (Plzf), octamer-binding transcription factor 4 (Oct-4), and Nanog [35]. Transcription factors such as the Myelocytomatosis oncogene (MYC) and E2F transcription factor 1 (E2F1) and kinases such as the protein kinase B (Akt) and death-associated protein kinase (DAPk) are impacted due to CIP2A regulation of PP2A leading to the inability of PP2A to dephosphorylate and thus deactivate these factors contributing to oncogenetic features of cells. Here, we will outline CIP2A’s role in multiple cellular responses, including the regulation of PP2A activity, immune responses, cell cycle and proliferation, senescence, EMT, extracellular matrix (ECM) turnover, and DNA repair.

### 2.1. CIP2A Regulation of PP2A Activity

PP2A has multiple functions but is primarily perceived as a tumor suppressor via dephosphorylating proteins that promote tumor growth [36]. Therefore, the regulation of PP2A by CIP2A is crucial in cancer initiation and progression. CIP2A is an obligatory homodimer that interacts with the PP2A regulatory subunit B′ isoform (B56) proteins of PP2A [37]. CIP2A directly binds to the PP2A-B56α trimer and displaces the PP2A-A subunit [38]. In this way, CIP2A competes with the B56α substrate binding by blocking the LxxIxE-motif substrate binding pocket on B56α [38]. Stabilizing PP2A and thus rendering CIP2A non-functional is a significant avenue to cancer therapies [39]. The kinases Akt and DAPk also contribute to the oncogenetic strength of CIP2A. Both Akt and DAPk are negatively regulated by PP2A [40]. Akt is a proto-oncogenic signaling pathway that is involved in cell proliferation and resistance to apoptosis. PP2A expression is associated with downregulation of the Akt pathway, where CIP2A overexpression is associated with Akt upregulation and cancer proliferation [41]. DAPk forms a complex with PP2A to create a CIP2A–PP2A– unc-5 homolog B (UNC5H2)–DAPK1 protein complex that helps eliminate tumor cells. CIP2A prevents this complex protein formation, creating an anti-apoptotic environment for cancer cells to proliferate [42]. Additional reading on PP2A-specific signaling in lung diseases can be found in the following review articles [43,44].

### 2.2. CIP2A Regulation of MYC Stabilization

MYC is the most reported CIP2A-regulated PP2A target. However, CIP2A can regulate the serine/threonine phosphorylation of approximately 100 target proteins in many different pathological and physiological processes [45]. CIP2A stabilizes the MYC protein which can lead to cell proliferation, invasion, and migration [1]. CIP2A also promotes the expression and nuclear export of p27Kip1 (Cyclin-dependent kinase inhibitor 1B; a cyclin-dependent kinase inhibitor), leading to its inactivation and the release of cell cycle arrest [46]. High levels of MYC contribute to the protooncogenic properties of CIP2A [47]. In gastric and breast cancers, increased MYC expression with concurrent CIP2A expression correlate with more resilient cancer cells [47,48]. CIP2A deletion is associated with decreased MYC expression, and destabilization of MYC also alters CIP2A expression [47]. CIP2A impacts Akt signaling, mechanistic target of rapamycin (mTOR) C1, E2F1, polo-like kinase 1 (Plk1), and microRNAs, all of which contribute to CIP2A’s ability to contribute to malignancies [49].

### 2.3. CIP2A’s Role in Cell Cycle and Senescence

CIP2A plays a role in multiple phases of the cell cycle. Its activity is particularly important in the G1/S transition, S phase, and G2/M transition. CIP2A promotes the transition from G1 to S phase by stabilizing MYC and enhancing its transcriptional activity [50]. MYC induces the expression of cyclin D1 and cyclin E, which activate cyclin-dependent kinase (CDK) 4/6 and CDK2 to phosphorylate retinoblastoma protein, thereby releasing E2F and allowing transcription of genes required for DNA synthesis [51]. CIP2A regulates the functions of CDK1 and CDK2 to facilitate the G1/S transition [52]. CIP2A also inhibits PP2A-mediated dephosphorylation of Akt, sustaining Akt signaling, which promotes cell survival and progression through G1 [53]. CIP2A is involved in the S Phase by stabilizing MYC, thus stabilizing the expression of genes involved in DNA replication and nucleotide biosynthesis [54]. This MYC and Akt signaling promote DNA replication and prevent replication stress-induced apoptosis [54].

CIP2A influences the G2 phase and G2/M transition by enhancing the activity of PLK1 and Aurora B kinase by preventing their dephosphorylation by PP2A [55]. This enhanced activity of PLK1 and Aurora B are essential for mitotic spindle formation, chromosome alignment, and mitotic entry. CIP2A also promotes the phosphorylation and activation of CDC25C, a phosphatase that activates CDK1 that is required for mitotic entry [56]. CIP2A is involved in the M Phase of the cell cycle by ensuring proper spindle checkpoint function by maintaining Aurora B kinase activity [57]. It also helps prevent mitotic arrest by sustaining Akt and MYC signaling during mitosis.

Alternatively, loss of CIP2A expression leads to MYC destabilization, which can trigger a senescence-like growth arrest [48]. Therefore, overexpression of CIP2A reduces cellular senescence. E2F1 overexpression, due to p53 or p21 inactivation, promotes expression of CIP2A, which in turn increases the stabilization of serine 364 phosphorylation of E2F1 [48]. This E2F1-CIP2A feedback loop is a key determinant of breast cancer cell sensitivity to senescence induction. Induction of senescence, whether via p53 activation or direct adenoviral overexpression of p21, is entirely dependent on CIP2A expression [48]. Therefore, CIP2A plays multiple roles in the cell cycle and senescence.

### 2.4. Emerging Evidence of CIP2A Regulation of EMT in Cancerous and Non-Cancerous Cells

Depletion of CIP2A is observed to inhibit cell proliferation, migration, invasion, and EMT in vitro in multiple different cancer cell lines [58,59]. This is linked to CIP2A influencing H-Ras, mitogen-activated protein kinase (MEK)/ERK, Akt, β-catenin, zinc finger protein SNAI2 (Slug), zinc-finger transcription factor (Snail), MYC, and PP2A signaling [60,61,62]. Treatment with ethoxysanguinarine (a CIP2A inhibitor) reduced CIP2A expression in a 2,3-butanedione-induced rat model of bronchiolitis obliterans, which coincided with reduced EMT [4]. CIP2A regulation of EMT is linked to mesenchymal traits that enhance the invasiveness, migration, and metastatic potential in cancer cells and fibrotic and inflammation-associated responses in non-cancerous cells [50].

### 2.5. CIP2A Signaling Linked to Immune Responses

CIP2A can regulate several inflammatory processes, including NFκB signaling, macrophage polarization, oxidative stress, reactive oxygen species (ROS) production, T-cell activation, and IL (interleukin)-17 signaling. Knockdown of CIP2A expression suppresses NFκB signaling in a PP2A-dependent manner [12]. This inhibition of CIP2A responses coincides with reduced proliferation, migration, and invasion in osteosarcoma cells [12]. In an animal model of bronchiolitis obliterans, inhibition of CIP2A signaling was observed to reduced NFκB signaling, inflammatory cell infiltration into the lungs, and secretions of IL-1β, IL6, and TNFα [4].

Inhibiting CIP2A expression with the CIP2A inhibitor TD52 reduced the release of inflammatory cytokines and apoptosis in macrophages and promoted macrophage autophagy regulation possibly via Akt/mTOR inhibition in an acute pancreatitis model [63]. Another derivative of erlotinib, TD19, is reported to regulate inflammatory responses in orchitis by inhibiting CIP2A expression [64]. Overexpression of CIP2A in astrocytes promote the release of IL1α, IL6, and TNF α [65]. CIP2A can also promote oxidative stress, which amplifies inflammation by triggering pathways like mitogen-activated protein kinases (MAPK) and c-Jun N-terminal kinase (JNK) [66].

CIP2A is also involved in regulating T-cell activation, potentially affecting autoimmune responses in diseases like rheumatoid arthritis and lupus [67]. CIP2A expression negatively regulates Th17 cell differentiation, via its interactions with acylglycerol kinase (AGK) and signal transducer and activator of transcription 3 (STAT3) [68]. CIP2A-deficient mice display an impaired adaptive immune response when challenged with *Listeria monocytogenes*, with decreased frequency of both cluster of differentiation (CD) 4+ T-cells and CD8+ effector T-cells [67]. Therefore, CIP2A regulation of immune responses needs to be investigated in several infection conditions.

### 2.6. Evidence for CIP2A Regulation of TGFβ Signaling

There is some evidence linking CIP2A responses to TGFβ-signaling. TGFβ can inhibit expression of a CIP2A binding protein (CIP2A-BP) resulting in enhanced CIP2A functional signaling [8]. This CIP2A-BP is encoded within the long intergenic noncoding RNA 00665 (LINC00665) [8], whose regulation is altered in many cancers [69]. Reduced CIP2A-BP expression enhances the amount of free-CIP2A to bind to PP2A and inhibit PP2A responses. The serine/threonine kinase TGFβ-activated kinase-1 (TAK1) is regulated by CIP2A in a PP2A-dependent manner in multiple myeloma cells [70]. Silencing CIP2A expression or use of the CIP2A inhibitor TD52 suppressed TAK1 phosphorylation and induced cell death in these cells [70]. Our group also observed that the PP2A activator, ATUX-1215, can reduce fibrotic markers and pulmonary function restriction in a bleomycin mouse model of fibrosis [9]. Whether this is CIP2A-mediated is yet to be determined. There may be several ways that CIP2A regulates TGFβ signaling; however, more studies are required to examine CIP2A regulation of TGFβ.

### 2.7. Other CIP2A Functions

CIP2A can inhibit glycolysis and promote oxidative metabolism in NSCLC cells. CIP2A binds to pyruvate kinase M2 (PKM2), which results in the formation of PKM2 tetramer [71]. By doing this, CIP2A redirects PKM2 to the mitochondrion, leading to B-cell lymphoma 2 (Bcl2) phosphorylation [71]. This downstream action triggered by CIP2A produced stronger suppressive effects on NSCLC cells both in vitro and in vivo when CIP2A was inhibited [71]. Exploring the proteins predicted to interact with CIP2A suggests that CIP2A may play a role in other functions, including RNA metabolic processing and splicing, protein traffic, cytoskeleton regulation, and ubiquitin-mediated protein degradation processes [72]. This suggests that CIP2A’s role in the cell may be more complex than what is outlined in this review.

## 3. Regulation of CIP2A Gene Expression and Protein Stabilization

In non-cancerous cells, CIP2A has relatively low gene expression throughout the body, except in sperm cells [35], intestinal stem cells [73], and some populations of T cells [68]. However, CIP2A is highly expressed in many tumor types [74]. Within these tumors, enhanced expression of CIP2A is not due to mutations or genetic enhancement of CIP2A but through transcriptional overexpression or by increased protein stability [75]. Understanding what regulates CIP2A expression and its functional roles enhances our understanding of CIP2A signaling but also opens multiple avenues to target CIP2A. In this way, many compounds have been identified that indirectly downregulate CIP2A expression [44] by either inhibiting CIP2A transcription [76] or by causing proteasomal degradation or instability of the CIP2A protein [77] (see Section 5 for more detail on these compounds and their therapeutic potential).

### 3.1. CIP2A Transcriptional Regulation

CIP2A gene expression is reported to be regulated by certain transcription factors, kinases, microRNAs (miRNA), and other factors. Transcription factors, such as MYC, E2F1, STAT3, and ETS proto-oncogene transcription factor (ETS) family proteins, can promote CIP2A expression by binding to the CIP2A gene promoter [74]. CIP2A also regulates signaling for several of these transcription factors; for example, both MYC and CIP2A positively regulate the expression of each other with a feedback loop between the two proteins [47]. CIP2A also stabilizes the E2F1 protein [78], in addition to stimulating activating MAPK, and is in itself regulated by MAPK [79]. The activated transcription factor (ATF2) can bind to the AP-1 site in the promoter region of CIP2A to initiate gene transcription in mouse embryonic fibroblasts [80]. Equally, the transcription factor CP2 (TFCP2) can bind to the CIP2A promoter to regulate its expression [81].

Signaling associated with DNA damage can promote CIP2A expression via phosphorylation and activation of the checkpoint kinase 1 (CHK1) [82,83]. CIP2A expression also correlates with mutations in the tumor-suppressor gene *TP53* (p53 protein), and suppressing p53 expression triggers CIP2A mRNA expression [48]. When p53 or p21 are inactivated, E2F1 becomes overexpressed resulting in the expression of CIP2A, stabilizing the serine 364 phosphorylation of E2F1 [84]. Histone deacetylase 1 (HDAC1) can regulate CIP2A expression by modifying chromatin structure, with inhibition of HDAC1 leading to decreased CIP2A transcription [85].

CIP2A expression is also regulated by miRNAs. Several miRNAs are known to negatively regulate CIP2A expression, such as miR-375 [86,87], miR-218 [88], miR-548b-3p [89], and miR-383-5p [90]. CIP2A is positively regulated by miR-301a [91]. Specificity protein 1 (SP1)-induced long noncoding mRNA transcript of the developmental pluripotency-associated 2 gene (RNA-DPPA2) upstream binding RNA binds to miR-520d-5p to enhance CIP2A expression [92].

The other factors reported to regulate CIP2A are PI2K/Akt [93], Octamer-binding transcription factor 4 (Oct4) [94], ATF6 [95], and PLK1 [96]. IL-10 phosphorylates cyclic adenosine monophosphate (cAMP) response element-binding protein (CREB) through the phosphoinositide 3-kinase (PI3K)/Akt signaling pathway to enhance CIP2A gene expression [93]. 17β-estradiol stimulus leads to the activation of the epidermal growth factor receptor (EGFR) MEK1/2 and PI3K pathways that increases the expression of CIP2A, via ETS1 signaling [97]. Oct4 positively regulates the expression of CIP2A in embryonic stem cells and testicular cancer cell lines [94]. Endoplasmic reticulum stress-related ATF6 can also regulate CIP2A expression [95]. CIP2A prevents PLK1 degradation [98], but PLK1 inhibition increases CIP2A expression [96]. This may be due to the C-terminal tail of CIP2A triggering 14-3-3 binding [86]. Finally, one study observed the reduced expression of CIP2A when using PP2A activators, suggesting that PP2A activity may also trigger suppression of CIP2A expression [99].

### 3.2. CIP2A Protein Stability

There are several mechanisms known to stabilize or destabilize the CIP2A protein and alter its function. CIP2A protein stability can be influenced by interactions with other proteins, including PP2A subunits [37,100], and can be negatively regulated by certain long non-coding RNAs [8]. Signaling that alters CIP2A dimerization decreases CIP2A protein stability and function [37]. CIP2A is also stabilized when it interacts with the B56 family of PP2A subunits [37]. Metformin inhibits CIP2A responses by detaching CIP2A from the PP2A complex, which leads to CIP2A degradation [100]. TGFβ-mediated inhibition of the CIP2A-BP results in enhanced CIP2A functional signaling [8]. This CIP2A-BP is encoded within the long intergenic noncoding RNA 00665 (LINC00665) [8], whose regulation is altered in many cancers [69]. CIP2A interacts with the long noncoding RNA tumor protein p53 target gene 1 (TP53TG1) to trigger its ubiquitination-mediated degradation, resulting in reduced CIP2A response and the inhibition of the PI3K/Akt pathway [62].

CIP2A protein stability is also regulated by the C-terminus of Hsc-70 interacting protein (an E3 ubiquitin ligase)-mediated ubiquitination [77], and mutation of the most prevalently ubiquitinated lysine 647 on CIP2A stabilizes the protein [38]. CIP2A’s function is also regulated by its cellular localization and subsequent protein interactions. CIP2A is primarily located in the cytoplasm, but CIP2A colocalizes with MYC and TopBP1 in the inner nuclear membrane in interphase cells [73]. Figure 1 summarizes the major functions of CIP2A and factors that regulate its expression and stability.

## 4. CIP2A Expression in Pulmonary Diseases

CIP2A upregulation is linked to several signaling cascades associated with pulmonary diseases, such as excessive proliferation, inflammation, EMT, fibroblast activation, and collagen deposition. Therefore, CIP2A may play a role in other pulmonary diseases beyond what is outlined here.

### 4.1. Lung Cancer

CIP2A overexpression is observed in various subtypes of lung cancer, including NSCLC [6] and SCLC [5]. In NSCLC, CIP2A promotes tumor progression by inhibiting the tumor suppressor activity of PP2A, leading to sustained oncogenic signaling through the pathways outlined above, such as MYC, Akt, and mTOR [101]. Elevated CIP2A levels correlate with poor prognosis and increased resistance to a range of chemotherapeutic agents, including platinum-based drugs such as cisplatin and carboplatin, as well as taxanes like paclitaxel and docetaxel [102]. These agents are commonly used in lung cancer treatment, but CIP2A overexpression impairs their efficacy by enhancing the survival of signaling pathways and reducing apoptotic responses [102]. CIP2A enhances MYC-driven transcriptional programs, promoting cell proliferation and survival [73]. Additionally, CIP2A-mediated PP2A inhibition results in increased Akt phosphorylation, contributing to tumor growth and metastasis [103]. IL-10 mRNA expression in lung tumors positively correlates with CIP2A mRNA expression [93].

Recent studies have linked CIP2A expression to EMT in lung cancer, a process whereby epithelial cells lose their cell–cell adhesion properties and acquire a more migratory and invasive mesenchymal phenotype, which is crucial for tumor invasion and metastasis [104]. EMT is a key event in cancer progression, facilitating metastasis by allowing cancer cells to penetrate tissue barriers and disseminate to distant sites. CIP2A upregulation is associated with the altered expression of EMT markers such as N-cadherin and vimentin [60]. Recent publications also show CIP2A playing a role in EMT in non-cancerous conditions [4,8].

### 4.2. Chronic Obstructive Pulmonary Disease (COPD)

Enhanced CIP2A expression is elevated in smoke exposure conditions and in samples isolated from COPD patients [2,3]. Our research group has demonstrated enhanced expression of CIP2A in human bronchial epithelial (HBE) cells isolated from COPD subjects compared with HBE cells from nonsmokers without COPD [2]. CIP2A expression was induced by exposure to cigarette smoke in HBE cells and in mice. This is of interest, as reactive oxygen species (ROS) (e.g., hydrogen peroxide) can suppress CIP2A expression in cancer conditions [100]. However in non-cancerous conditions, chronic smoke induction of CIP2A coincided with a reduction in PP2A activity, airspace enlargements, and loss of lung function in mice [2]. Interestingly, modulating CIP2A expression in HBE cells by silencing RNA or chemically with erlotinib enhanced PP2A activity reduced extracellular-signal-regulated kinase phosphorylation and reduced the responses of matrix metalloproteinases (MMP) 1 and 9 in HBE cells [2]. Reactivation of PP2A in mice also reduced cigarette smoke-induced emphysema in mice [105,106,107]. A recent study also confirmed elevated CIP2A expression in combination with reduced PP2A responses in lung epithelial cells isolated from COPD subjects [3]. They reported that treatment with a caveolin-1 scaffolding domain peptide could reverse these CIP2A/PP2A responses and alter MMP-12 production, type II pneumocyte cell viability, and mucus production [3]. It is also of note that altered PP2A responses are observed in alpha-1 antitrypsin deficiency [106,108]; however, CIP2A expression has not been reported in this disease.

### 4.3. CIP2A Responses in Fibrosis

CIP2A overexpression is linked to abnormal tissue remodeling and fibrogenesis [8], but it remains unclear whether CIP2A overexpression is a driving factor initiating fibrotic changes or a downstream consequence of chronic tissue injury and remodeling processes. CIP2A may influence TGFβ receptor activity indirectly by altering the phosphorylation status of key signaling proteins. TGFβ can inhibit the expression of CIP2A-BP, resulting in enhanced CIP2A functional signaling, and enhances the amount of free-CIP2A to bind to PP2A and inhibit PP2A responses [8]. This CIP2A-BP is encoded within the long intergenic noncoding RNA 00665 (LINC00665) [8]. Expression of LINC00665 is higher in IPF lung tissue, and smoke exposure appears to further enhance this expression [109]. LINC00665 expression is upregulated in bleomycin-induced mouse lung fibrosis tissues, and inhibition of LINC00665 expression suppressed fibrogenesis in bleomycin-induced lung fibrosis [109]. Equally, our group and others have demonstrated that PP2A activity is reduced in IPF, and chemical activation of PP2A can reduce bleomycin-induced fibrosis in mice [9,110,111]. A recent study on cartilage degradation demonstrated that CIP2A regulates cartilage degeneration and inflammation by targeting the cell migration-inducing protein and NFκB responses [112]. This suggests that CIP2A would have a role in ECM responses. However, more research is needed to determine the direct role of CIP2A in fibrosis and IPF.

### 4.4. CIP2A Regulation in Other Pulmonary Diseases

In conditions such as asthma and allergic airway diseases, CIP2A may modulate immune responses by either directly influencing immune cells like T cells and macrophages or by altering upstream cytokine signaling pathways, which subsequently modulate their activity [67,68]. Most studies linking CIP2A to these conditions are due to its ability to regulate PP2A, and direct evidence demonstrating a role for CIP2A is required. Inhibition of PP2A enhances the activation of NF-κB and other pro-inflammatory transcription factors, leading to increased production of inflammatory mediators such as IL-4, IL-5, and IL-13 [113]. PP2Aα is downregulated in smooth muscle cells in asthma, and PP2Aα activation inhibits bronchoconstriction when examining bronchi isolated from mice [114]. CIP2A regulates the interaction between the acylglycerol kinase and STAT3 in Th17 cells, which modulates STAT3 phosphorylation and expression of IL-17 [68]. CIP2A-deficient mice display an impaired adaptive immune response when challenged with *Listeria monocytogenes*, with decreased frequency of both CD4+ T-cells and CD8+ effector T-cells [67]. Altered PP2A responses are associated with increased histamine release and enhanced allergic reactions [115,116]; however, the involvement of CIP2A in these processes requires further investigation. Equally, abnormal activity of PP2A causes corticosteroid insensitivity in severe asthma [117]. In two asthma mouse models, activation of PP2A reduced the severity of acute and chronic allergic airway disease by suppressing tissue eosinophils and inflammation, mucus-secreting cell numbers, IL-33, thymic stromal lymphopoietin, IL-5, IL-13, serum IgE, and airway hyper-responsiveness [118]. Despite PP2A being linked to asthma and allergy, CIP2A has not been directly studied in this context.

Finally, a recent article observed that CIP2A expression was enhanced in a 2,3-butanedione-induced rat model of bronchiolitis obliterans, and treatment with ethoxysanguinarine reduced the CIP2A expression, intraluminal occlusion, inflammatory infiltration, and fibrosis [4].

## 5. Therapeutic Targeting of CIP2A

There are several approaches to target CIP2A, directly and indirectly. This can be achieved by targeting CIP2A expression, inhibiting upstream CIP2A activity, CIP2A protein stability, or blocking its downstream primary targets, such as activating PP2A responses. Inhibiting CIP2A decreases cancerous cell viability and anchorage-independent growth and reduces the oncogenic potential of certain cells [1]. CIP2A deficiency in mouse models demonstrates no growth abnormalities [48]. Similarly, rescuing PP2A activity should induce a therapeutic response in cancer cells [119]. RNA interference (RNAi) may also represent a feasible approach, as CIP2A RNAi treatment inhibited tumor growth in urothelial carcinoma in vitro [120]. Since overexpression of CIP2A can upregulate the drug resistance of tumor cells to chemotherapy [92], it is plausible that suppressing CIP2A signaling may make tumors that were resistant to some chemotherapies sensitive to this treatment again. Therefore, effective therapeutic responses against cancer cells may be most effective with both inhibition of kinase signaling pathways (such as CIP2A and its signaling) and the reactivation of downstream signaling, such as PP2A [49]. Here, we outline the available and experimental options to target CIP2A expression and signaling (See Table 1).

### 5.1. Targeting CIP2A Expression

Erlotinib, a food and drug administration (FDA)-approved EGFR kinase inhibitor for treating EGFR-mutant NSCLC, can reduce CIP2A levels, which indirectly restores the tumor-suppressive activity of PP2A [121]. Equally, erlotinib inhibits CIP2A expression in bronchial epithelial cells from COPD subjects [2]. However, it is important to note that the pharmacokinetics of erlotinib is different in current smokers and nonsmokers, with increased metabolic clearance of erlotinib observed in current smokers [122]. Therefore, smoking habits must be taken into consideration when utilizing erlotinib for treatment. It is also reported that erlotinib can target CIP2A independent of EGFR mutations [123]. There are several experimental erlotinib derivatives known to target CIP2A, including TD-19 and TD-52 [124,125]. Tumors that are resistant to erlotinib are reported to be TD-19-sensitive [124]. Lapatinib, a tyrosine kinase inhibitor primarily utilized to treat human epidermal growth factor receptor 2 (HER2)-positive advanced or metastatic breast cancer, was observed to induce significant apoptosis and inhibit CIP2A and p-Akt in triple-negative breast cancer cell lines by blocking CIP2A signaling [126].

Temsirolimus, an FDA-approved drug for advanced renal cell carcinoma (RCC), suppresses CIP2A transcription and promotes its degradation through the lysosomal–autophagy pathway, disrupting the CIP2A-mTOR interaction and reducing phosphorylation of ERK and Akt, particularly in non-Kirsten rat sarcoma viral oncogene homolog (K-Ras) mutant cell lines [127]. Clinical trials have examined the use of Temsirolimus for the treatment of NSCLC [63,128]. Lapatinib, a dual tyrosine kinase inhibitor used for ErbB2-positive breast cancer, inhibits CIP2A, resulting in the inactivation of Akt signaling and increased apoptosis in cancer cells [129].

The expression of CIP2A is decreased following treatment with the small molecular compounds that activate PP2A, ATUX-792, and DBK-1154 in multiple cancer cell lines, without altering the expression of another endogenous PP2A inhibitor, SE translocation (SET) [99]. It would be of interest to assess these compounds in terms of CIP2A expression in non-cancerous cells or tissue. We and others have confirmed increased PP2A activity in the lungs of mice administered ATUX-792 and DBK-1154 and a diarylmethyl-pyran-sulfonamide compound called ATUX-1215 [9,105,106]. Administration of these compounds slowed the progression of smoke-induced COPD, alpha-1 antitrypsin deficiency associated emphysema, and bleomycin-induced fibrosis in mouse models [9,105,106].

### 5.2. Targeting CIP2A Protein for Inactivation or Degradation

Several compounds are known to alter CIP2A protein levels. Treatment with the proteasome inhibitor bortezomib effectively lowered CIP2A levels, reactivating PP2A and suppressing Akt signaling [130]. Additionally, this study suggested that combining CIP2A-targeting therapies with other inhibitors could mitigate oncogene addiction by shifting dependency from EGFR to Akt signaling pathways [130]. Penfluridol (PF), an FDA-approved antipsychotic drug, has demonstrated potential for treating melanoma and its metastases, particularly in the brain and lungs, by targeting CIP2A [131]. Unlike many drugs, PF crosses the blood–brain barrier, addressing a significant challenge in treating melanoma brain metastases. PF enhances the interaction between CIP2A and its E3 ligase, von Hippel–Lindau (VHL), leading to CIP2A degradation via the ubiquitin–proteasome pathway. This reactivates PP2A, suppresses Akt and MYC, and significantly reduces melanoma cell viability, inducing apoptosis, and inhibiting migration and invasion [131].

Several naturally derived compounds, such as celastrol, gambogenic acid (GEA), and tenuigenin promote CIP2A degradation. Celastrol (tripterine), derived from traditional Chinese medicine, promotes CIP2A degradation and thereby inhibits its function [77]. Similarly, ethoxysanguinarine, a benzophenanthridine alkaloid extracted from *Macleaya cordata*, can inhibit CIP2A expression and activity and subsequently downregulate MYC and Akt responses [132]. GEA, a natural compound derived from *Garcinia hanburyi*, targets CIP2A [133] by inducing CIP2A degradation via the ubiquitin–proteasome pathway, disrupting oncogenic signaling pathways like MYC and phosphorylated Akt and enhances the sensitivity of hepatocellular carcinoma cells to anticancer agents [133]. Furthermore, GEA can synergistically augment bortezomib-induced apoptosis in multiple myeloma cells [133,134]. Tenuigenin, a known NFκB inhibitor isolated from the root of the Chinese herb *Polygala tenuifolia*, can inhibit CIP2A expression and signaling [12] and significantly attenuated *Staphylococcus aureus*-induced lung histopathological changes and inflammatory cytokines TNF-α and IL-1β production in an animal model [135].

Metformin, a biguanide antidiabetic drug, is recommended as the first-line therapy for type 2 diabetes due to its efficacy, relative safety, and beneficial effects of reducing hemoglobin A1c (HbA1C) levels and weight, in addition to its general tolerability and favorable cost [136]. Interestingly, metformin can increase PP2A activity by inhibiting CIP2A responses [137]. Several reports suggest that metformin has potential therapeutic benefits for COPD [138] and lung cancer [139]. Subjects on metformin treatment have reduced rates of exacerbations and hospitalizations, along with a slower decline in lung function [140]. A systematic review and meta-analysis found that metformin use in patients with diabetes and COPD reduced the risk of COPD-related hospitalizations and showed a trend toward lower all-cause mortality without increasing the risk of hyperlactatemia [141]. Similarly, a meta-analysis of 18 studies demonstrated that metformin is significantly associated with a decreased risk of developing lung cancer and increased survival in lung cancer patients [139].

### 5.3. Indirect Targeting CIP2A Signaling

There are several studies that outline reduced CIP2A responses that may not be directly CIP2A gene- or protein-regulated. Fingolimod (FTY720), an immunomodulatory drug for multiple sclerosis, demonstrates anticancer potential by activating PP2A and potentially inhibiting CIP2A activity [142]. Additionally, combining PP2A activators or CIP2A inhibitors with kinase inhibitors (e.g., MEK inhibitors) or chemotherapy enhances anticancer effects, particularly in aggressive cancers like lung cancer [143]. Other endogenous PP2A regulators, such as PME-1 and SET, are also being explored as therapeutic targets, with compounds like FTY720 showing the potential to restore PP2A function while inhibiting CIP2A expression [144]. While compounds such as bortezomib (FDA-approved for multiple myeloma) and Fingolimod (FDA-approved for multiple sclerosis) have shown potential to modulate PP2A or CIP2A activity in preclinical studies, their use specifically for targeting CIP2A in cancer remains experimental. There is some evidence that the traditional Chinese medicine Xuebijing, which comprises a combination of five herbal extracts, can alter PP2A responses [145], but thus far, there is no link to CIP2A expression. A recent meta-analysis study suggests that Xuebijing may increase the oxygenation index, lower the respiratory rate, and improve acute physiology and chronic health evaluation II (APACHE II) scores and inflammatory biomarkers in acute lung injury and ARDs patients [146]. Further studies are required to validate these approaches. See Table 1 for a summary of potential CIP2A inhibitors.

**Table 1 medicina-61-01740-t001:** Potential approaches to target CIP2A signaling.

Drug/Compound	Approval Status	Mode of Action	Therapeutic Relevance
Erlotinib (Tarceva^®^)	FDA-approved (NSCLC)	Reduces CIP2A levels, restoring PP2A activity, via inhibition of EGFR signaling [119]	Enhances sensitivity to EGFR-targeted therapies, overcoming resistance in NSCLC
Erlotinib derivatives, e.g., TD-19 and TD-52	Experimental	Downregulation of CIP2A expression and p-AKT expression and increased PP2A activity; reduced EGFR involvement [124,125]	Inhibit CIP2A expression independent of EGFR
Bortezomib (Velcade^®^)	FDA-approved (Multiple Myeloma)	Downregulates CIP2A expression, reducing Akt activity [133,134]	Promotes apoptosis and tumor growth inhibition in hematologic cancers
Lapatinib (Tykerb^®^)	FDA-approved (Breast Cancer)	Inhibits CIP2A, leading to Akt inactivation [126]	Reduces tumor proliferation in ErbB2-positive breast cancer
Afatinib (Gilotrif)	FDA-approved (NSCLC)	Reduces Elk-1 binding to the CIP2A promoter and suppresses CIP2A expression [76]	Reduces tumor cell proliferation and triggers cell death, especially in tumors with EGFR mutations
Fingolimod (FTY720)	FDA-approved (Multiple Sclerosis)	Activates PP2A, potentially modulating CIP2A indirectly [144]	Shows promise in enhancing anticancer signaling and reducing inflammation
Celastrol	Experimental	Reduces CIP2A stability via proteasomal degradation [77]	Demonstrates antiproliferative effects in preclinical cancer models
FTY720 derivatives (e.g., FTY720-C2, FTY720-Mitoxy, FTY720 vinylphosphonate, and FTY720 methyl ether)	Experimental	Disrupt SET-PP2A complex, reducing CIP2A activity	Enhance PP2A activation, showing promise in preclinical cancer studies
Metformin	FDA-approved (Diabetes)	Reduces CIP2A levels through proteasomal degradation [137]	Potential application in neurodegenerative diseases and cancers with CIP2A involvement
Penfluridol (PF)	FDA-approved (Antipsychotic)	Directly targets CIP2A, promoting its degradation via the ubiquitin-proteasome pathway by enhancing CIP2A interaction with its E3 ligase VHL [131]	Shows potential in treating melanoma and metastases, particularly in the brain and lungs, by reactivating PP2A and disrupting oncogenic pathways like Akt and MYC
Temsirolimus + Cetuximab	FDA-approved (Temsirolimus: RCC, Cetuximab: EGFR+ cancers)	Temsirolimus suppresses CIP2A transcription and promotes its degradation via lysosomal autophagy; Cetuximab blocks EGFR activation, inhibiting downstream oncogenic signaling [127]	Synergistic effects demonstrated in colon cancer models, reducing tumor growth, inducing apoptosis, and improving patient outcomes by targeting CIP2A and EGFR pathways
Gambogenic acid	Experimental	Induces CIP2A degradation via the ubiquitin-proteasome pathway, inhibiting MYC and p-Akt signaling [133]	Enhances sensitivity to anticancer agents in hepatocellular carcinoma and potentiates bortezomib-induced apoptosis in multiple myeloma cells
Ethoxysanguinarine	Experimental	Inhibits CIP2A activity, disrupting oncogenic signaling pathways associated with CIP2A overexpression [4]	Derived from *Macleaya cordata*, shows potential as a CIP2A-targeting agent in preclinical studies for cancer treatment
Small molecular activators of PP2A, e.g., ATUX-792, DBK-1154, and ATUX-1215	Experimental	Inhibit CIP2A expression in cancer cell lines; bind directly to PP2A, but the mechanism for the inhibition of CIP2A expression is unknown [99]	Potential utility in diseases where CIP2A is increased and PP2A activity is subdued
Tenuigenin	Experimental	Inhibits via CIP2A and NFκB signaling, via PP2A [12]	A bioactive ingredient from *Polygala tenuifolia*; has potential to reduce NFκB signaling and inflammation
Cucurbitacin B	Experimental	Promotes the lysosomal degradation of EGFR and the downregulation of the CIP2A/Akt and activation of PP2A [147]	A natural tetracyclic triterpenoid compound mainly found in Cucurbitaceae

## 6. Systemic Targeting of CIP2A Responses

The previous sections explored the potential benefits of targeting CIP2A, particularly in the context of pulmonary diseases. However, as with any therapeutic strategy, it is important to consider the possible limitations and side effects of CIP2A inhibition or secondary effects of these treatments. This section discusses the potential risks, including the systemic effects, immune system modulation, and the tolerability of current FDA-approved CIP2A inhibitors.

### 6.1. Limitations and Risks of Targeting CIP2A

As already mentioned, the cells reported to express CIP2A in normal homeostasis conditions are sperm cells, intestinal stem cells, and some immune cells [35]. CIP2A is expressed in the testis where it plays a role in sperm production. Studies in mice deficient for CIP2A have a marked decrease in sperm production and significantly smaller and lighter epididymides [35]. While reversing CIP2A inhibition restored sperm production, these findings raise concerns about the potential for infertility in men receiving CIP2A inhibitors, especially those planning to have children.

In cancerous conditions, CIP2A plays a role in regulating the cell cycle by inhibiting the G2/M check point thus allowing damaged cells to progress through mitosis [74]. CIP2A also plays a major role in DNA repair pathways, particularly with TopB1. CIP2A also controls phosphorylation and mitosis, which can affect DNA repair mechanisms, especially in BRCA mutant cancer cells, which lead to genomic instability [148]. However, the role of CIP2A mitosis and DNA repair in non-cancerous cells and tissues needs to be explored in greater detail.

CIP2A is involved in immune modulation, but CIP2AHOZ (CIP2A deficient) mice appear to have normal immune system development and function. However, when challenged with *Listeria monocytogenes*, these mice have a decreased frequency of both CD4+ T-cells and CD8+ effector T-cells [67]. This suggests that CIP2A might play a role in T-cell-mediated activation [67]. Inhibiting CIP2A expression with TD52 and ethoxysanguinarine decreased the release of inflammatory cytokines, reduced macrophage apoptosis, promoted macrophage autophagy regulation, and inhibited the Akt-mTOR pathway in an in vitro model of acute pancreatitis [63]. Therefore, immune modulation may be observed in CIP2A-deficient or inhibition conditions.

Equally, prolonged inhibition of CIP2A could result in PP2A hyperactivation, which would likely impact the cell cycle progression, apoptosis, DNA repair, signal transduction (e.g., Akt, MYC, MAPK pathways), inflammation, and other signaling processes. A recent study linked activation of PP2A in endothelial cells through the formation of a complex involving apoER2, Disabled-2 (Dab2), and Src homology domain-containing transforming protein 1 (SHC1) [149]. This resulted in the dephosphorylation and inhibition of endothelial nitric oxide synthase (eNOS) and Akt, resulting in reduced nitric oxide production and subsequently promoting a prothrombotic state [149]. However, this has not been reported in any additional studies, and the role of CIP2A in thrombosis has not been investigated.

Finally, PP2A plays a role in other factors, such as neurological effects (PP2A is crucial for maintaining normal brain function and cognitive dysfunction), fertility, and inflammation suppression. There is also some evidence suggesting that CIP2A may play a role in neurodegenerative diseases such as Parkinson’s Disease [150]. In a study of patients with Parkinson’s disease, lower levels of CIP2A were observed. This could indicate a potential link between CIP2A inhibition and neurodegeneration. However, the data are limited, and the exact causality of CIP2A inhibition on such effects is unclear.

### 6.2. Tolerability of CIP2A Inhibitors

The clinical tolerability of CIP2A inhibitors is a critical consideration when evaluating their use in lung disease treatment. Several CIP2A inhibitors, including Erlotinib, Lapatinib, Afatinib, Bortezomib, Niclosamide, Fingolimod, Metformin, Penfluridol, Temsirolimus, and Cetuximab have been extensively studied for their effectiveness and side effects. The diseases targeted with these drugs and the common and possible serious side effects are summarized in Table 2. Currently, there is no evidence to suggest that CIP2A is a direct cause of Erlotinib, Lapatinib, Afatinib, Bortezomib, Niclosamide, Fingolimod, Metformin, Penfluridol, Temsirolimus, and Cetuximab’s common and most frequently reported side effects. CIP2A could possibly impact the side effects of these compounds in an indirect manner outside of their primary targets (e.g., EGFR, mTOR, and Akt). However, this requires further investigation.

## 7. Future Perspectives

Whether CIP2A functions as an endogenous inhibitor of PP2A to modulate phosphorylation-dependent pro-inflammatory, pro-fibrotic, and pro-proliferative pathways in the lung requires further exploration and discussion. This regulation of PP2A responses is likely a major means why CIP2A signaling is linked to different pulmonary pathologies. As outlined above, preclinical evidence supports both CIP2A inhibition and PP2A reactivation as plausible therapeutic strategies, but clinical translation requires more human data and drug toxicity investigations, especially in non-cancerous conditions. Functional analyses of CIP2A in all lung cell types and detailed expression pattern analysis in human clinical samples are areas requiring comprehensive studies. Equally, validation studies utilizing multiple animal models of disease would further our knowledge of the potential causality of CIP2A expression in these diseases. Currently, there is a lack of knowledge of CIP2A expression profiles, and functional data are needed in multiple cell types. The pulmonary field has little information on which lung cell types (e.g., epithelial subtypes, fibroblast subsets, or alveolar macrophages) rely most on CIP2A expression in disease. The cell-specific roles of CIP2A need to be addressed with single cell sequencing or spatial sequencing throughout healthy and diseased lungs. While PP2A dysfunction is documented in lung cancer, COPD, and IPF, direct large-scale human data linking elevated CIP2A expression to clinical fibrosis severity are still limited. Equally, the prolonged activation of PP2A or inhibition of CIP2A needs extensive testing for efficacy and safety. Systemic inhibition of CIP2A and subsequent long-term activation of PP2A could have off-target effects. Targeted delivery or selective modulation of CIP2A–PP2A interactions may be preferable. Directly inhibiting CIP2A or reactivation of PP2A has the potential to restore homeostasis, resulting in the dephosphorylation of pro-pathogenic substrates (MYC, Akt, NFκB, and SMADs). Equally, directly targeting NFκB may regulate the CIP2-PP2A associated signaling linked to altered EMT, inflammation, and cell proliferation. CIP2A regulation of proliferation is likely due to MYC stabilization and activation of Akt/ERK responses. CIP2A regulation of ECM remodeling and EMT may be due to modulation of TGFβ/SMAD and NFκB responses. Equally, modulation of NFκB would also alter immune responses. However, substantial clinical data are needed prior to conducting possible future clinical trials targeting CIP2A and PP2A responses.

Finally, there is no evidence to suggest that targeting CIP2A expression would contribute to the side effects observed with current FDA-approved compounds. However, this requires further investigation. Equally, modulating CIP2A expression may enhance cancer cell susceptibility to chemotherapy resistant tumors, thereby allowing for enhanced responses to well established cancer treatments.

## 8. Conclusions

CIP2A expression is linked to several observations observed in pulmonary diseases, including inflammation (increased activation of NFκB, MAPKs, and pro-inflammatory cytokines), tissue remodeling (enhanced TGF-β signaling and fibroblast activation), and PP2A signaling inhibition (sustained inhibition of PP2A, resulting in the activation of oncogenic and pro-inflammatory pathways). Understanding these mechanisms is crucial for developing targeted therapies aimed at modulating CIP2A activity in pulmonary diseases. Future research should focus on identifying specific inhibitors of CIP2A and assessing their therapeutic potential in various lung conditions. CIP2A’s role in normal cell function appears to be less significant compared to its role in cancer cells, suggesting that targeting CIP2A signaling may be a safe approach for therapy. However, more investigations are needed in non-cancerous environments.

## Figures and Tables

**Figure 1 medicina-61-01740-f001:**
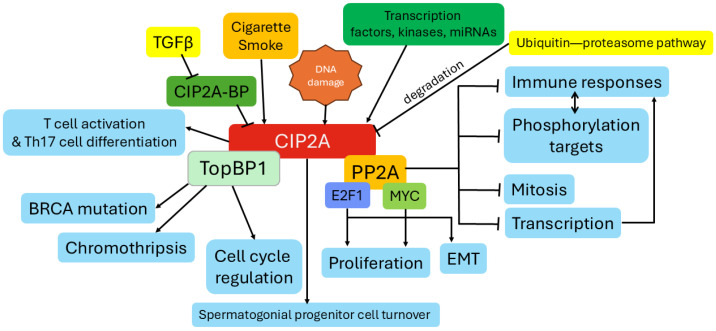
Regulation and functional role of CIP2A.

**Table 2 medicina-61-01740-t002:** Systemic effects of FDA approved drugs that target CIP2A signaling.

Drug/Compound	Approved for the Treatment of	Reported Side Effects
Erlotinib (Tarceva^®^)	Metastatic NSCLC; advanced pancreatic cancer	*Common side effects:* rash, dry and itching skin, thinning hair, brittle and inflamed nails, diarrhea, nausea, vomiting, loss of appetite, weight loss, mouth sores, cough, shortness of breath, fatigue, headache, numbness or tingling in hands and feet, bone or muscle pain, eye irritation, redness, or dryness
Bortezomib (Velcade^®^)	Multiple Myeloma; Mantle Cell Lymphoma	*Common side effects:* peripheral neuropathy, nausea, vomiting, diarrhea, constipation, fatigue, low blood cell counts, and rash *Serious side effects:* heart and lung complications and development of posterior reversible encephalopathy syndrome
Lapatinib (Tykerb^®^)	HER2-positive advanced or metastatic breast cancer	*Common side effects:* diarrhea, hand–foot syndrome, nausea, vomiting, fatigue, and rash *Serious side effects:* interstitial lung disease development and cardiovascular complications
Afatinib (Gilotrif)	Metastatic NSCLC; metastatic squamous NSCLC	*Common side effects:* diarrhea and rash
Fingolimod (FTY720)	Relapsing forms of multiple sclerosis (MS), such as clinically isolated syndrome, relapsing–remitting disease, active secondary progressive disease	*Common side effects:* headache, changes in liver function tests, and increased susceptibility to infections *Serious side effects:* Progressive Multifocal Leukoencephalopathy, macular edema, liver injury, and potential worsening of MS symptoms
Metformin	Type 2 diabetes	*Common side effects:* diarrhea, nausea, gas, stomach pain, fatigue, and weight loss *Serious side effects:* lactic acidosis, vitamin B12 deficiency, and hypoglycemia
Penfluridol	Chronic schizophrenia and other related psychotic disorders	*Common side effects:* dizziness, restlessness, sedation, weight gain, gastrointestinal issues, tremors, and rigidity
Temsirolimus	Advanced renal cell carcinoma	*Common side effects:* rash, pimples, dry skin, skin blemishes, mouth irritation or sores, constipation, diarrhea, stomach pain, upset stomach, decreased appetite, fatigue, headache, back pain, muscle or joint pain, difficulty sleeping, changes in taste, weight loss, nosebleeds, runny or stuffy nose, and throat irritation *Serious side effects:* hives, rash, itching, difficulty breathing or swallowing, flushing, chest pain, shortness of breath, abdominal pain, abdominal swelling, chills, fever, constipation, nausea, vomiting, increased infection, fever, sore throat, chills, cough, nausea, confusion, dizziness or faintness, weakness or numbness of an arm or leg, red blood in stools, and decrease in the amount of urine
Cetuximab	Colorectal and head and neck cancers	*Common side effects:* skin rash, dry skin, itching, and nail issues *Serious side effects:* infusion reactions, heart complications, and electrolyte imbalances

## Data Availability

Not applicable.

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
