# Peer review of "The Emerging Role of the Cancerous Inhibitor of Protein Phosphatase 2A in Pulmonary Diseases"

_medicina, 2025, doi:10.3390/medicina61101740_

Round 1
Reviewer 1 Report
Comments and Suggestions for Authors
Thank you for the opportunity to review this valuable manuscript on the role of CIP2A in pulmonary diseases. This paper is highly timely and important, as it comprehensively covers the role of CIP2A—a protein gaining attention in oncology—not only in lung cancer but also in non-cancerous pulmonary conditions like COPD and pulmonary fibrosis. The structure is logical and clear, and the figures and tables greatly aid the reader's understanding.
While the manuscript is very well-written overall, I believe a few revisions and additions could further enhance its value.
Overall Assessment
This is an excellent review that elucidates the multifaceted role of the oncoprotein CIP2A across a wide range of pulmonary diseases. The organization of information regarding potential therapeutics targeting CIP2A and their associated risks is particularly beneficial for both basic researchers and clinicians. I strongly recommend this manuscript for publication in this journal after revisions have been addressed.
Specific Suggestions for Improvement
Major Points
- Deepen the Comparative Discussion on CIP2A's Role Across Different Pulmonary Diseases; The manuscript covers multiple diseases, including lung cancer, COPD, and pulmonary fibrosis. I suggest further deepening the discussion on the functional commonalities and differences of CIP2A among these conditions. A more explicit authorial perspective in the "Discussion" or "Conclusion" section on how CIP2A is involved—either commonly or specifically—in core pathological processes like inflammation, fibrosis, and cell proliferation would significantly enhance the review's originality.
- Provide Specific Recommendations for Future Research; As the manuscript suggests, many aspects of CIP2A's role in COPD and pulmonary fibrosis remain to be elucidated. To help guide future studies, it would be beneficial to add a section such as "Future Perspectives." This section could propose specific research questions, such as functional analyses of CIP2A in specific cell types, validation using animal models of disease, or detailed expression pattern analysis in human clinical samples. This would provide readers with a clear roadmap for future investigations.
Minor Points
- Suggestion to Improve Figure 1; Figure 1 provides an excellent summary of CIP2A's function and regulation. To improve clarity, I recommend differentiating the arrows in the signaling pathways to indicate "activation" (→) and "inhibition" (T-bar). This would allow readers to grasp the relationships between the elements more intuitively.
- Elaborate on the Side Effects of Therapeutics; Table 2, which summarizes the side effects of existing drugs with CIP2A inhibitory effects, is very informative. In the section "6. Systemic targeting of CIP2A responses," could you add a brief discussion on whether these side effects are thought to be caused by the inhibition of CIP2A itself or by the primary mechanism of each drug (e.g., EGFR inhibition)? This perspective is crucial for assessing the true risks of targeting CIP2A.
- Unification of Abbreviations; The abbreviations are well-defined throughout the manuscript. However, a final check to ensure that all abbreviations are defined with their full terms upon first use would be appreciated.
Conclusion
This manuscript is a valuable and clear review of the relationship between CIP2A and pulmonary diseases. I am confident that its quality will be further enhanced by addressing the minor points raised above.
Author Response
Reviewer 1:
Comment 1: Deepen the Comparative Discussion on CIP2A's Role Across Different Pulmonary Diseases; The manuscript covers multiple diseases, including lung cancer, COPD, and pulmonary fibrosis. I suggest further deepening the discussion on the functional commonalities and differences of CIP2A among these conditions. A more explicit authorial perspective in the "Discussion" or "Conclusion" section on how CIP2A is involved—either commonly or specifically—in core pathological processes like inflammation, fibrosis, and cell proliferation would significantly enhance the review's originality.
Response 1: We have tried to address this comment and the subsequent comment into a discussion and future perspectives section. Please see lines 599-634 of the article with track changes
Comment 2: Provide Specific Recommendations for Future Research; As the manuscript suggests, many aspects of CIP2A's role in COPD and pulmonary fibrosis remain to be elucidated. To help guide future studies, it would be beneficial to add a section such as "Future Perspectives." This section could propose specific research questions, such as functional analyses of CIP2A in specific cell types, validation using animal models of disease, or detailed expression pattern analysis in human clinical samples. This would provide readers with a clear roadmap for future investigations.
Response 2: We thank the reviewer for their constructive recommendations. Please see lines 599-634 of the article with track changes
Comment 3: Suggestion to Improve Figure 1; Figure 1 provides an excellent summary of CIP2A's function and regulation. To improve clarity, I recommend differentiating the arrows in the signaling pathways to indicate "activation" (→) and "inhibition" (T-bar). This would allow readers to grasp the relationships between the elements more intuitively.
Response 3: We have made minor adjustments as recommended to the figure.
Comment 4: Elaborate on the Side Effects of Therapeutics; Table 2, which summarizes the side effects of existing drugs with CIP2A inhibitory effects, is very informative. In the section "6. Systemic targeting of CIP2A responses," could you add a brief discussion on whether these side effects are thought to be caused by the inhibition of CIP2A itself or by the primary mechanism of each drug (e.g., EGFR inhibition)? This perspective is crucial for assessing the true risks of targeting CIP2A.
Response 4: So far there is limited data to suggest that inhibition of CIP2A is contributing to the side effects of these compounds. We have noted this on lines 590-595 of the article with track changes
Comment 5: Unification of Abbreviations; The abbreviations are well-defined throughout the manuscript. However, a final check to ensure that all abbreviations are defined with their full terms upon first use would be appreciated.
Response 5: We have followed your suggestion and have now defined all abbreviations. Please see updates throughout the manuscript and the abbreviation table on line 664 of the article with track changes
Comment 6: This manuscript is a valuable and clear review of the relationship between CIP2A and pulmonary diseases. I am confident that its quality will be further enhanced by addressing the minor points raised above.
Response 6: We thank the reviewer for their helpful suggestions. We have made all the recommended changes to the manuscript.
Reviewer 2 Report
Comments and Suggestions for Authors
The present review article “The emerging role of the cancerous inhibitor of protein phos-2 phatase 2A in pulmonary diseases” describes the role of CIP2A in pulmonary diseases, highlighting it signaling pathways, regulatory mechanisms, and therapeutic potential as a target for novel treatments.
Strengths
- Comprehensive coverage of direct, indirect, and experimental therapeutic strategies.
- Good use of FDA-approved drugs (erlotinib, temsirolimus, lapatinib, metformin, bortezomib, fingolimod) to highlight potential drug repurposing.
- Table 1 reference is excellent — gives readers a concise summary of compounds.
Few comments to be addressed:
- Capitalize the P of promising in Abstract
- Subsection numbering (e.g., 2 comes after 2.3) is disordered.
- Transitions between sections are abrupt; the narrative lacks smooth transition between the topics e.g. often jumps between cancer-related CIP2A functions and pulmonary disease without smooth linking.
- CIP2A’s role in PP2A inhibition, MYC stabilization, and Akt signaling is explained multiple times in slightly different ways
- Background on CRD global burden is detailed, but repeated in parts and distracts from the main CIP2A focus.
- Figure 1 is helpful but still busy. A clearer figure with grouped pathways (oncogenic vs immune vs fibrotic) would make the review more readable.
- Citations are not consistent (currently mixed with bracketed numbers like [1], [2-5], [15,16]).
- Reference numbers sometimes appear out of order. Example: [47] is cited before [43,44].
Please add the following reference related to your review
- Bin, Y., Peng, R., Lee, Y., Lee, Z., & Liu, Y. (2025). Efficacy of Xuebijing injection on pulmonary ventilation improvement in acute pancreatitis: a systematic review and meta-analysis. Frontiers in Pharmacology, 16, 1549419. doi: 10.3389/fphar.2025.1549419
- Kang, L., Gao, X., Liu, H., Men, X., Wu, H., Cui, P.,... Yan, J. (2018). Structure–activity relationship investigation of coumarin–chalcone hybrids with diverse side-chains as acetylcholinesterase and butyrylcholinesterase inhibitors. Molecular Diversity, 22(4), 893-906. doi: 10.1007/s11030-018-9839-y
Author Response
Reviewer 2:
Comment 1: Comprehensive coverage of direct, indirect, and experimental therapeutic strategies. Good use of FDA-approved drugs (erlotinib, temsirolimus, lapatinib, metformin, bortezomib, fingolimod) to highlight potential drug repurposing. Table 1 reference is excellent — gives readers a concise summary of compounds.
Response 1: We thank the reviewer for their helpful suggestions
Comment 2: Capitalize the P of promising in Abstract
Response 2: We have corrected this typo
Comment 3: Subsection numbering (e.g., 2 comes after 2.3) is disordered.
Response 3: We have double-checked the numbering as suggested.
Comment 4: Transitions between sections are abrupt; the narrative lacks smooth transition between the topics e.g. often jumps between cancer-related CIP2A functions and pulmonary disease without smooth linking.
Response 4: We attempted to make transitions less abrupt. Unfortunately, there is limited data on CIP2A in non-cancerous conditions, which makes it difficult to the separation of cancerous vs non-cancerous roles of CIP2A.
Comment 5: CIP2A’s role in PP2A inhibition, MYC stabilization, and Akt signaling is explained multiple times in slightly different ways
Response 5: We have edited the document to reduce repetition but the majority of CIP2A studies examine these three proteins, and they have major roles in many processes.
Comment 6: Background on CRD global burden is detailed, but repeated in parts and distracts from the main CIP2A focus.
Response 6: We have edited the document to reduce repetition. Please see track changes in the document
Comment 7: Figure 1 is helpful but still busy. A clearer figure with grouped pathways (oncogenic vs immune vs fibrotic) would make the review more readable.
Response 7: We agree with the reviewer. However, most of these pathways will overlap in oncogenic, immune and fibrotic responses, which make it difficult to accurately subdivide into groups. We have made minor changes to the figure.
Comment 8: Citations are not consistent (currently mixed with bracketed numbers like [1], [2-5], [15,16]). Reference numbers sometimes appear out of order. Example: [47] is cited before [43,44].
Response 8: We are using the journals recommended EndNote style. But we will work with the journals’ editors to adjust the references to their preference.
Comment 9: Please add the following reference related to your review
Bin, Y., Peng, R., Lee, Y., Lee, Z., & Liu, Y. (2025). Efficacy of Xuebijing injection on pulmonary ventilation improvement in acute pancreatitis: a systematic review and meta-analysis. Frontiers in Pharmacology, 16, 1549419. doi: 10.3389/fphar.2025.1549419
Kang, L., Gao, X., Liu, H., Men, X., Wu, H., Cui, P.,... Yan, J. (2018). Structure–activity relationship investigation of coumarin–chalcone hybrids with diverse side-chains as acetylcholinesterase and butyrylcholinesterase inhibitors. Molecular Diversity, 22(4), 893-906. doi: 10.1007/s11030-018-9839-y
Response 9: We thank the reviewer for their helpful suggestions. We have incorporated the recommendation for the first reference to the manuscript. However, we could not find any evidence that coumarin or chalcones directly impacted CIP2A responses. This may be something worth investigating. However, due to the lack of evidence we have omitted the second reference currently. Please see lines 520-528 of the article with track changes